# Preclinical Evaluation of the Systemic Safety, Efficacy, and Biodistribution of a Recombinant AAV8 Vector Expressing FIX-TripleL in Hemophilia B Mice: Implications for Human Gene Therapy

**DOI:** 10.3390/ijms26136073

**Published:** 2025-06-24

**Authors:** Sheng-Chieh Chou, Cheng-Po Huang, Ying-Hui Su, Chih-Hsiang Yu, Yung-Li Yang, Ssu-Chia Wang, Yi-Hsiu Lin, Yen-Ting Chen, Jia-Yi Li, Yen-Ting Chang, Su-Yu Chen, Shu-Wha Lin

**Affiliations:** 1Department of Internal Medicine, National Taiwan University Hospital, College of Medicine, National Taiwan University, Taipei 100, Taiwan; choushengchieh@ntuh.gov.tw; 2Trineo Biotechnology Co., Ltd., New Taipei City 221, Taiwan; cphuang1978@gmail.com (C.-P.H.); wangminbd@gmail.com (S.-C.W.); r03424003@ntu.edu.tw (Y.-H.L.); ytc2018ix.hbd@gmail.com (Y.-T.C.); lijiare@gmail.com (J.-Y.L.); 3Childhood Cancer Foundation, Taipei 100, Taiwan; r89424018@ntu.edu.tw; 4Institute of Statistical Science Academia Sinica, Taipei 115, Taiwan; yarguespirit@gmail.com; 5Department of Clinical Laboratory Sciences and Medical Biotechnology, National Taiwan University, Taipei 100, Taiwan; winnie86110148@gmail.com; 6Department of Pediatrics, National Taiwan University Hospital, Taipei 100, Taiwan; yangyl92@ntu.edu.tw; 7Department of Laboratory Medicine, National Taiwan University Hospital, Taipei 100, Taiwan; 8Department of Laboratory Medicine, College of Medicine, National Taiwan University, Taipei 100, Taiwan

**Keywords:** gene therapy, hemophilia B, AAV8, FIX, FIX-TripleL

## Abstract

Gene therapy for hemophilia B offers the advantage of a single administration with sustained therapeutic effects. This study evaluated the systemic safety, efficacy, biodistribution, and immunogenicity of AAV8-FIX-TripleL, a recombinant adeno-associated virus type 8 (AAV8) vector encoding a modified factor IX (FIX) variant with increased activity. In this good laboratory practice (GLP)-compliant study, 180 male FIX-knockout hemophilia B mice were randomized into 12 groups (*n* = 15) and received intravenous AAV8-FIX-TripleL at therapeutic (5 × 10^11^ VG/kg) or supraphysiological (5 × 10^12^ VG/kg) doses on Day 1. The mice were sacrificed on Days 2, 15, 28, and 91 for comprehensive evaluations, including hematological and biochemical assessments, histopathological examination, FIX protein/activity analysis, immunogenicity assessment, and vector biodistribution via quantitative polymerase chain reaction (qPCR) in major organs. AAV8-FIX-TripleL demonstrated dose-dependent increases in FIX activity and protein levels, with FIX activity exceeding physiological levels and the maintenance of a favorable safety profile. Biodistribution analysis confirmed predominant hepatic accumulation and vector persistence up to 91 days post-injection, with minimal off-target distribution. These findings indicate that AAV8-FIX-TripleL is a promising gene therapy candidate for hemophilia B, as it has robust expression, sustained efficacy, and a favorable safety profile, and that further translational studies are warranted.

## 1. Introduction

Mutations in the F8 and F9 genes cause the X-linked bleeding disorder categorized as hemophilia A and B, respectively [1,2]. Patients with severe hemophilia A or B suffer from frequent spontaneous bleeding in joints, muscles, or other organs and eventually experience permanent damage or even death if the disorder is not properly treated [3]. The current standard treatment for severe hemophilia is the prophylactic replacement of coagulation factors via the intravenous route [4,5], which should be initiated as early as possible and continue for life if no anti-factor inhibitors develop. Hemophilia B is indistinguishable from hemophilia A through clinical presentation alone [6]. Although intravenous replacement of factor IX (FIX) is generally given less frequently than that of factor VIII, and the risk of developing an anti-FIX inhibitor is substantially lower than the risk of developing an anti-factor VIII inhibitor, lifelong replacement of FIX is quite an undertaking for patients, caregivers, healthcare professionals, and the healthcare system [3,4,7,8].

Gene therapy, which involves transferring a functional copy of a defective gene to specific cells, is a direct treatment and possibly a cure for many genetic diseases, including hemophilia [9,10]. In 2012, Nathwani et al. made a breakthrough by inducing a FIX level of 1–6% via gene therapy for severe hemophilia B patients, and this level was sustained for over a decade. This was the first long-term sustainable gene therapy for hemophilia [11,12]. This success was accomplished through the use of a liver trophic AAV8 vector [13], optimized design of the F9 construct [14,15], and the administration of steroids to some patients with T-cell reactions to hepatocytes. Although there are still concerns and unknowns [16], to date, two gene therapies for hemophilia B, fidanacogene elaparvovec [17] and etranacogene dezaparvovec [18], have demonstrated excellent safety profiles and efficacy in clinical trials and have been approved in several countries. Both therapies are based on the same hyperfunctional variant of FIX [19,20], FIX-R338L, also known as FIX-Padua, which has approximately 8 times the clotting activity of regular FIX. Compared with the unmodified F9 construct, applying the hyperfunctional F9 variant enables much higher efficacy and potentially a lower dosage. A lower dosage of gene therapy might diminish the risk of adverse events, such as increases in transaminase levels and infusion reactions. Theoretically, a more hyperfunctional F9 variant would be even more suitable for gene therapy applications.

This study evaluated the preclinical safety, efficacy, biodistribution, thrombosis risk factors, and immunogenicity of AAV8-FIX-TripleL, a novel AAV8 vector encoding FIX-TripleL. This variant incorporates V86A, E277A, and R338L substitutions, conferring approximately 2.9-fold higher specific activity than FIX-R338L [21]. By leveraging the hepatotropic properties of AAV8 and utilizing a liver-specific ApoE/AAT promoter, AAV8-FIX-TripleL aims to achieve robust and sustained FIX expression while maintaining a favorable safety profile. Since hemophilia B affects primarily males [2], male F9-knockout mice were used as an animal model in this study. The dosing strategy was designed on the basis of prior preclinical studies involving AAV8-FIX-R338L, with a therapeutic dose of 10^11^ VG/kg and a supraphysiological dose of 10^12^ VG/kg [22].

In this good laboratory practice (GLP)-compliant study, the systemic safety, efficacy, biodistribution, thrombosis risk factors, and immunogenicity of AAV8-FIX-TripleL in FIX-knockout hemophilia B mice were systematically assessed, and essential data are provided to support the translational potential of AAV8-FIX-TripleL for clinical development.

## 2. Results

### 2.1. Evaluation of FIX Activity and Protein Expression Following AAV8-FIX-TripleL Administration in Hemophilia B Mice

Hemophilia B mice were administered AAV8-FIX-TripleL at doses of 5 × 10^11^ VG/kg (therapeutic dose) and 5 × 10^12^ VG/kg (supraphysiological dose), whereas the control groups received Dulbecco’s phosphate-buffered saline (DPBS). A summary of the treatment groups is provided in Table 1. Plasma samples were collected on Days 2, 15, 29, and 91 to assess FIX activity and protein expression (Figure 1). In the control groups, FIX protein and activity were undetectable. On Day 2, FIX activity was 33.6 ± 15.3% in the 5 × 10^11^ VG/kg group and 194.0 ± 38.8% in the 5 × 10^12^ VG/kg group, and protein concentrations were not detected. By Day 15, FIX activity increased to 832.6 ± 183.9% and 1349.2 ± 361.1%, with protein concentrations of 1857 ± 506 ng/mL and 16,591 ± 5673 ng/mL, respectively. The activity peaked on Day 29 (1149.7 ± 256.7% and 1784.4 ± 514.4%), while the protein levels reached 1584 ± 510 ng/mL and 17,113 ± 2425 ng/mL, respectively. By Day 91, the activity declined slightly to 890.9 ± 151.9% and 995.1 ± 265.7%, with protein concentrations of 2661 ± 695 ng/mL and 31,265 ± 7787 ng/mL, respectively. FIX activity and protein levels exhibited a dose-dependent trend. The specific activity peaked on Day 29 at 7878 ± 2546 U/mg in the 5 × 10^11^ VG/kg group, whereas the levels on Day 91 (3560 ± 1156 U/mg) were comparable to those on Day 15 (4677 ± 1007 U/mg). In the supraphysiological dose group, the specific activity was lower than that in the therapeutic dose group, likely owing to the saturation of assay reagents and interference between FIX in extremely high concentrations. FIX activity and protein expression were sustained for at least 90 days post-administration, with activity consistently above 100%. In addition, in a separate non-GLP study, we observed that the administration of a subtherapeutic dose (2.5 × 10^10^ VG/kg) resulted in a mean FIX activity of 30.7 ± 8.2% on Day 29 (Appendix A), which meets clinical therapeutic standards. This indicates that AAV8-FIX-TripleL exhibits promising efficacy even at low doses. Furthermore, in a long-term observation, mice administered a therapeutic dose (5 × 10^11^ VG/kg) maintained a mean FIX activity of 595.6 ± 158.2% one year post-injection (Appendix A), demonstrating sustained FIX expression and therapeutic effect over an extended period.

### 2.2. Biodistribution of AAV8-FIX-TripleL in Hemophilia B Mice

The biodistribution of AAV8-FIX-TripleL across seven organs (heart, liver, spleen, lung, kidney, brain, and testis) at Days 2, 15, 29, and 91 is shown in Figure 2. Consistent with the known hepatic tropism of AAV8, the highest vector DNA concentrations were observed in the liver in a dose-dependent manner. In the therapeutic dose group (5 × 10^11^ VG/kg), the liver vector DNA copy numbers were 3071.68 (Day 2), 1113.93 (Day 15), 3455.3 (Day 29), and 1565.6 (Day 91), whereas the levels in other organs remained minimal or undetectable. Similarly, in the supraphysiological dose group (5 × 10^12^ VG/kg), the liver copy numbers were 37,691.68 (Day 2), 15,990.13 (Day 15), 45,826.79 (Day 29), and 18,354.3 (Day 91). At both dose levels, the highest vector DNA levels outside the liver were observed in the spleen and heart on Day 2; however, these levels decreased by Days 15 and 29 and became undetectable by Day 91. At the supraphysiological dose, vector DNA remained detectable on Day 91 in the liver (15/15 mice), heart (9/15 mice), and brain (1/15 mice). However, the signals in the heart and brain were approximately 100-fold lower than those in the liver. The detection rates of vector DNA are summarized in Appendix A. At the therapeutic dose, vector DNA was exclusively detected in the liver (15/15 mice) on Days 29 and 91, with no detectable signal in other organs. A similar trend was observed in the supraphysiological dose group, where vector DNA was consistently detected in the liver (15/15 mice) at all time points. Moreover, the detection rates in other organs gradually declined and eventually became undetectable. These findings further confirm the strong hepatic tropism of AAV8-FIX-TripleL.

### 2.3. Safety Evaluation of AAV8-FIX-TripleL in Hemophilia B Mice

No animal deaths occurred throughout the study, and no apparent clinical signs were observed in the experimental groups. The mean body weights of the animals, summarized in Appendix A, were not significantly different between the AAV8-FIX-TripleL-treated mice and the control mice. Similarly, feed consumption remained consistent across all groups, indicating that there were no treatment-related adverse effects on general health. Gross necropsy revealed occurrences of diffuse subcutaneous hemorrhage in hemophilia B mice, a characteristic sign of the disease. Specifically, subcutaneous hemorrhage was observed in 2/15 (G1), 2/15 (G2), and 3/15 (G3) mice on Day 2; in 1/15 (G4) mice on Day 15; and in 1/15 (G10) mice on Day 91. Notably, in mice receiving therapeutic or supraphysiological doses of AAV8-FIX-TripleL, no subcutaneous hemorrhage was observed at any time point on Days 15, 29, or 91, suggesting effective amelioration of spontaneous bleeding events following treatment (Appendix A).

Hematology, biochemistry, and urinalysis results indicated that no pathological changes were attributable to AAV8-FIX-TripleL administration. Given the well-documented hepatotropism of AAV8, which can induce transient elevations in circulating liver enzymes [23], we assessed key hepatic biomarkers, including aspartate aminotransferase (AST), alanine aminotransferase (ALT), alkaline phosphatase (ALP), and lactate dehydrogenase (LDH), as presented in Figure 3. Notably, LDH serves as a marker not only for liver injury but also for cardiac and muscular damage [24,25]. Across all time points—Day 2, Day 15 (short-term), Day 29, and day 91 (long-term)—there were no statistically significant differences in the levels of these enzymes between the AAV8-FIX-TripleL-treated mice and the control mice. These findings indicate the absence of detectable liver damage following AAV8-FIX-TripleL administration.

Histopathological analysis, summarized in Table 2, revealed spontaneous lesions in multiple organs, including the lungs, heart, liver, and kidneys, and these lesions were present in both the AAV8-FIX-TripleL-treated and control mice (Appendix A). The severity of these lesions was evaluated and compared between groups, with no statistically significant differences observed on Days 2, 15, 29, or 91. These findings suggest that the observed histopathological changes were independent of AAV8-FIX-TripleL treatment. Additional histopathological findings were noted sporadically, including lesions in the pancreas and spleen on Day 2 (G1 and G2) and in the testis on Day 29 (G7 and G8). These findings occurred at a low frequency in both the control and AAV8-FIX-TripleL-treated groups, but their correlation with AAV8-FIX-TripleL remains uncertain. In terms of hepatic pathology, minimal necrosis was observed in the control group (2/15) on Day 15, whereas mild fibrosis was detected in one mouse (1/15) in the supraphysiological dose group on Day 29. However, statistical analysis revealed no significant differences between the AAV8-FIX-TripleL-treated and control groups, suggesting that these hepatic lesions were not related to treatment. Collectively, these findings indicate that AAV8-FIX-TripleL did not induce treatment-associated histopathological abnormalities and that the observed spontaneous lesions were of minimal to moderate severity, occurring independently of AAV8-FIX-TripleL administration.

### 2.4. AAV8-FIX-TripleL Does Not Increase the Levels of the Prothrombotic Markers Thrombin-Antithrombin Complex and D-Dimers

Hypercoagulation is a known risk factor for thrombosis, making its evaluation crucial for assessing the safety of AAV8-FIX-TripleL. The levels of the thrombosis risk indicators thrombin–antithrombin complex (TAT) and D-dimers [26,27] were measured, and the results are presented in Figure 4. No significant differences in TAT (Figure 4A) or D-dimer levels (Figure 4B) were observed across the treatment groups. These findings suggest that intravenous administration of AAV8-FIX-TripleL at both therapeutic and supraphysiological doses does not lead to a significant increase in thrombosis risk at doses up to 5 × 10^12^ VG/kg.

### 2.5. Immunogenicity Evaluation of AAV8-FIX-TripleL in Hemophilia B Mice

To assess the immunogenicity of AAV8-FIX-TripleL, we measured the plasma levels of anti-AAV8 IgG and anti-FIX IgG antibodies, as well as those of Bethesda inhibitors (Table 3). The enzyme-linked immunosorbent assay (ELISA) results revealed that anti-AAV8 IgG was undetectable on Day 2 but became detectable in all AAV8-FIX-TripleL-treated mice on Days 15, 29, and 91, with mean values ranging from 5.21 to 42.50 μg/mL. Anti-FIX IgG was detected in a single mouse on Day 15 (5 × 10^11^ VG/kg) and in multiple mice (*n* = 4) on both Days 29 and 91 in the supraphysiological dose group (5 × 10^12^ VG/kg), with levels of 7.61, 3.60, and 0.57 μg/mL, respectively. No Bethesda inhibitors were detected in any treatment group. These findings indicate that AAV8-FIX-TripleL induced an adaptive immune response against the AAV8 capsid, but the anti-FIX IgG levels were minimal, and no neutralizing antibodies were observed.

## 3. Discussion

This study evaluated the FIX coagulation activity, biodistribution, and immunogenicity of AAV8-FIX-TripleL in a hemophilia B mouse model and provided critical insights into the therapeutic potential of AAV8-FIX-TripleL. AAV8-FIX-TripleL incorporates the TripleL variant, which exhibits increased specific activity due to three point mutations (V86A/E277A/R338L) [21]. The construct is driven by a liver-specific promoter (ApoE/AAT) to ensure targeted hepatic expression, whereas AAV8, known for its strong hepatic tropism [23], was selected as the vector to facilitate efficient FIX delivery. The goal of this design is to achieve sustained FIX activity with a single administration. Our findings demonstrate that AAV8-FIX-TripleL enables early and sustained FIX expression. FIX activity was detectable as early as Day 2 post-injection, with the therapeutic dose (5 × 10^11^ VG/kg) achieving a peak in FIX activity of 1149.7% on Day 29 and maintaining levels above 800% through Day 91. Additionally, the average FIX protein concentration remained above 1500 ng/mL from Day 15 to Day 91, indicating stable and prolonged FIX expression, a key factor for long-term therapeutic efficacy.

Biodistribution analysis confirmed that AAV8-FIX-TripleL was predominantly localized in the liver, which is consistent with the well-established hepatic tropism of AAV8 vectors [22,28]. Notably, viral vector DNA levels in the liver exceeded 1500 copies/mg on Day 90, demonstrating prolonged vector persistence. In contrast, vector DNA levels in non-hepatic tissues declined significantly over time, suggesting a favorable distribution pattern that minimizes off-target effects and increase safety. The restricted hepatic expression of AAV8-FIX-TripleL underscores its potential clinical utility in hemophilia B therapy.

Necropsy revealed subcutaneous hemorrhage in all groups on Day 2. However, by Days 15, 29, and 91, no subcutaneous bleeding was observed in the mice that received either the therapeutic dose or the supraphysiological dose. Although additional pharmacodynamic assessments of coagulation beyond FIX activity were not included in this study, these observations suggest that AAV8-FIX-TripleL administration effectively reduces the occurrence of spontaneous bleeding episodes, characteristic of hemophilia B. Histopathological analysis revealed immune cell infiltration in the lung, heart, and liver across multiple groups, including the control group, indicating that these findings were not associated with AAV8-FIX-TripleL treatment. Furthermore, even at the supraphysiological dose, no significant liver damage was observed (Table 2). The absence of hepatic toxicity was further supported by biochemical analysis (Figure 3), which corroborated the histopathological findings. These results provide critical evidence that AAV8-FIX-TripleL has a favorable safety profile, even at high doses.

Physiological FIX activity is typically maintained within the range of 50–150% [29]. In this study, no thrombotic events or adverse clinical manifestations were observed even at a supraphysiological dose (5 × 10^12^ VG/kg), where peak FIX activity reached 1784%. Furthermore, levels of coagulation biomarkers such as TAT and D-dimer remained comparable to those in the control groups, suggesting that homeostatic mechanisms, including FIX saturation or negative feedback loops, may regulate FIX activity. While no overt thrombotic risk was identified, long-term studies are warranted to assess the potential for hypercoagulability, particularly in high-dose settings.

AAV8-FIX-TripleL exhibited significantly greater specific activity than existing FIX gene therapy variants, including FIX-Padua. In our dosing strategy, which is based on prior studies [22], 10^11^ VG/kg was used as a reference; however, our results indicate that FIX activity exceeds 100%. In contrast, clinical trials of AAV8-FIX have demonstrated that FIX activity levels of 5–40% are sufficient to significantly reduce the risk of bleeding episodes [17,18]. Although no increase in thrombotic risk was observed, the long-term impact of excessively high FIX activity on coagulation balance should be evaluated in future studies. These findings suggest that the high expression efficiency of FIX-TripleL may necessitate dose optimization to lower levels (e.g., 10^10^ VG/kg) to achieve therapeutic efficacy while reducing potential thrombotic risk and immunogenicity.

Immunogenicity analysis revealed that all AAV8-FIX-TripleL-treated mice developed anti-AAV8 IgG antibodies in a dose- and time-dependent manner, which is consistent with previous reports on AAV-mediated gene therapies [28,30]. This response suggests that AAV-based delivery may limit further readminister [31]. Anti-FIX IgG antibodies were primarily observed in the high-dose group (5 × 10^12^ VG/kg), but their titers declined over time, indicating that the immune response against FIX-TripleL was transient and did not result in persistently high titers. Importantly, no Bethesda inhibitors were detected, supporting the potential for sustained efficacy without significant neutralizing antibody formation against FIX.

The U.S. Food and Drug Administration (FDA) has approved fidanacogene elaparvovec and etranacogene dezaparvovec as gene therapies for hemophilia B [17,18]. These therapies are based on the FIX-R338L (Padua) variant under the control of a liver-specific promoter. The therapeutic success of fidanacogene elaparvovec is attributed to its favorable safety profile at supraphysiological doses, sustained liver-specific expression, and prolonged maintenance of the activity of FIX in the circulation [17,32,33]. Similarly, in other gene therapies, such as verbrinacogene setparvovec, the FIX-R338L variant is also incorporated [34]. In contrast, AAV8-FIX-TripleL uniquely employs the FIX-TripleL variant, exhibiting 2.9-fold higher specific activity than FIX-R338L [21]. In our study, no thrombotic risk was observed in animals up to 90 days post-administration, even at supraphysiological doses. Furthermore, long-term expression analysis demonstrated that in mice receiving a 5 × 10^11^ VG/kg dose, FIX activity remained at an average of 596% one year post-injection, indicating stable and sustained expression. These findings highlight the potential advantages of AAV8-FIX-TripleL, including its high specific activity, liver-targeted expression, and prolonged therapeutic efficacy, supporting its promise as a next-generation gene therapy candidate for hemophilia B.

Overall, this study provides preclinical evidence supporting the therapeutic potential of AAV8-FIX-TripleL, highlighting its safety profile, efficacy, biodistribution, and immunogenicity, all of which are critical factors for its advancement toward clinical application.

## 4. Materials and Methods

### 4.1. Recombinant AAV Vectors

GLP-grade AAV8-FIX-TripleL vectors were produced and characterized at TFBS Bioscience, Inc., New Taipei City, Taiwan. Quality control assessments, including visual inspection, bioburden testing, endotoxin level measurement, purity analysis, and potency evaluation, were conducted by the company (Appendix A). This vector encodes FIX-TripleL, a human FIX variant (FIX-V86A/E277A/R338L) previously developed by our group [21], which exhibits 15-fold greater specific activity than wild-type FIX (FIX-WT). The viral genome consists of 7218 nucleotides of single-stranded DNA flanked by AAV2 inverted terminal repeats (ITRs). The genome includes an ApoE-HCR enhancer, a human α1-antitrypsin (hAAT) promoter, the FIX-TripleL coding sequence, and a bovine growth hormone (bGH) polyadenylation signal. AAV8-FIX-TripleL is a recombinant, replication-defective, nonenveloped AAV serotype 8 vector produced via triple plasmid cotransfection in HEK293T cells. This process involves (1) an ITR-containing plasmid carrying the FIX-TripleL gene, (2) a plasmid encoding adenoviral helper genes, and (3) a plasmid providing AAV Rep-Cap functions.

### 4.2. Animal Experiments

Hemophilia B mice, specifically FIX-knockout (KO) mice, were derived from the B6.129P2-F9tm1Dws strain and backcrossed with C57BL/6 mice for more than 10 generations [21]. Given that hemophilia B predominantly affects males, this study utilized male F9-knockout mice as an animal model. A total of 180 male mice were randomly assigned to 12 groups (*n* = 15 per group) on the basis of sacrifice time points and dosing regimens. The groups included controls, a therapeutic dose (5 × 10^11^ VG/kg), and a supraphysiological dose (5 × 10^12^ VG/kg), with scheduled sacrifices on Days 2, 15, 29, and 91. Following gas anesthesia (isoflurane/oxygen), orbital blood was collected into sodium citrate (3.2% sodium citrate-to-blood ratio of 1:9)-containing tubes, K2EDTA-containing tubes, and Eppendorf tubes for subsequent separation of plasma, white blood cells/reticulocytes, and serum, respectively. The FIX activity, FIX protein, TAT, D-dimer levels, anti-AAV8, anti-FIX IgG, and FIX neutralizing antibody titers of the plasma samples were analyzed. Samples in K2EDTA-containing tubes were used for hematological analysis, while serum from Eppendorf tubes was used for biochemical assessments. All animal procedures were approved by the Institutional Animal Care and Use Committee (IACUC-2021-SH-002, approved on 19 March 2021; IACUC-2020-SH-013, approved on 25 May 2020) of Trineo Biotechnology Co. (New Taipei City, Taiwan).

### 4.3. ELISA for Quantitating hFIX in Plasma

Plasma human FIX (hFIX) concentrations were measured using a commercially available ELISA kit (Assaypro LLC, St. Charles, MO, USA) following the manufacturer’s instructions. Briefly, plasma samples were diluted and added to 96-well plates precoated with a FIX-specific antibody. Bound hFIX was detected using a peroxidase-conjugated anti-human FIX monoclonal antibody. After washing, a colorimetric reaction was initiated by adding a hydrogen peroxide substrate solution, and the reaction was stopped with an acidic solution. The absorbance was measured at 450 nm, with a reference reading at 570 nm, via a multimode microplate reader (CLARIOstar, BMG, Ortenberg, Germany). The corrected absorbance (450–570 nm) was directly proportional to the hFIX protein concentration, which was quantified using a standard curve generated from known hFIX concentrations.

### 4.4. FIX Activity Assay

FIX activity was measured using an activated partial thromboplastin time (aPTT) assay [21,35] on a fully automated coagulation analyzer (CA-600 series, Sysmex, Hyogo, Japan). Briefly, plasma samples were diluted with CA system buffer (Siemens, Munich, Germany) containing 0.1% bovine serum albumin (BSA) and mixed with human FIX-deficient plasma. Coagulation was initiated by adding Actin^®^ FSL activator (Siemens, Munich, Germany) and calcium chloride (CaCl_2_), and the clotting time was recorded. A standard curve was generated through the use of serially diluted human standard plasma (Siemens, Munich, Germany), with activity values assigned according to the manufacturer’s Certificate of Analysis (C.O.A.). The FIX activity in the experimental samples was calculated from the standard curve on the basis of the recorded aPTT. The FIX activity of pooled human standard plasma was defined as 100% activity, equivalent to 1 unit of FIX per milliliter (U/mL). Specific FIX activity, expressed as units of FIX activity per milligram of protein (U/mg), was determined by dividing FIX activity (U/mL) by the FIX protein concentration (mg/mL).

### 4.5. Detection of hFIX DNA in Organs

Tissue homogenization was performed using a homogenizer (SpeedMill Plus, Analytik Jena, Jena, Germany). Genomic DNA was extracted from homogenized heart, lung, liver, spleen, kidney, brain, and reproductive organs using the DNA Mini Kit 480 (LabTurbo, Taipei, Taiwan) in combination with the Automated Nucleic Acid Purification System (LabTurbo, Taipei, Taiwan), following the manufacturer’s instructions. DNA concentrations were subsequently measured using a microvolume spectrometer (NanoVue Plus, GE HealthCare, Chicago, IL, USA) or a microvolume spectrophotometer (Analytik Jena, Jena, Germany). hFIX DNA was quantified via real-time quantitative PCR (qPCR) by using AAV8-FIX-TripleL-specific primers. This method was modified from a previous study on the biodistribution of AAV gene therapy formulations [36]. Briefly, 10 ng of tissue DNA was mixed with primers and SYBR Green mix, followed by amplification using a real-time PCR system (StepOnePlus, Applied Biosystems, Waltham, MA, USA). A standard curve was generated using serially diluted AAV8-FIX-TripleL plasmid DNA (TFBS Bioscience, Inc., New Taipei City, Taiwan). The copy number of hFIX DNA in each sample was then calculated from the standard curve and expressed as copies per 10 ng of genomic DNA. The sequences of primers used were as follows: forward (F): 5′-AACCAGCAGTGCCATTTC-3′ and reverse (R): 5′-CATCTTCTCCACCAACAACC-3′.

### 4.6. Measurements of TAT and D-Dimer Concentrations

Plasma TAT concentrations were quantified using a commercially available Mouse TAT ELISA Kit (Assaypro LLC, St. Charles, MO, USA). Briefly, diluted plasma samples were added to 96-well plates precoated with a TAT-specific antibody. The bound TAT was detected using a peroxidase-conjugated anti-TAT monoclonal antibody. After a series of washing steps, a colorimetric reaction was initiated by adding a hydrogen peroxide substrate solution and subsequently stopped with an acidic solution. The absorbance was measured at 450 nm with a reference at 570 nm via a multimode microplate reader (CLARIOstar, BMG, Ortenberg, Germany). The corrected absorbance (450–570 nm) was directly proportional to the TAT concentration, which was determined using a standard curve generated from known TAT concentrations. Plasma D-dimer concentrations were measured using a Mouse D-dimer Enzyme Immunoassay Kit (Cloud-Clone Corp., Katy, TX, USA) in a competitive inhibition ELISA format. Diluted plasma samples and biotin-labeled D-dimer were added to 96-well plates precoated with a D-dimer-specific monoclonal antibody. After incubation, peroxidase-labeled avidin was added, followed by washing steps. A colorimetric reaction was initiated with a hydrogen peroxide substrate solution and stopped with an acidic solution. The absorbance was measured at 450 nm using a multimode microplate reader (CLARIOstar, BMG, Ortenberg, Germany). The D-dimer concentration in each sample was calculated using a standard curve, with the signal intensity inversely proportional to the D-dimer levels.

### 4.7. Detection of the Anti-AAV8 Capsid IgG Antibody

Anti-AAV8 capsid IgG antibody concentrations were measured using ELISA. Briefly, 96-well ELISA plates precoated with AAV8-FIX-TripleL viral particles were used to capture anti-AAV8 capsid IgG antibodies from diluted plasma samples. Bound antibodies were detected using a biotin-labeled anti-mouse IgG monoclonal antibody, followed by streptavidin-peroxidase. After a series of washing steps, a colorimetric reaction was initiated by adding a hydrogen peroxide substrate solution and subsequently stopped with an acidic solution. The absorbance was measured at 450 nm with a reference at 650 nm via a multimode microplate reader (CLARIOstar, BMG, Ortenberg, Germany). The corrected absorbance (450–650 nm) was directly proportional to the anti-AAV8 capsid IgG antibody concentration, which was determined using a standard curve generated from known concentrations of the anti-AAV8 (intact particle) mouse monoclonal antibody (Progen, Heidelberg, Germany).

### 4.8. Detection of Anti-FIX IgG Antibodies

Anti-FIX IgG antibody concentrations were determined using an ELISA-based assay. Briefly, 96-well ELISA plates precoated with hFIX native protein (Invitrogen, Waltham, MA, USA) were used to capture anti-FIX IgG antibodies from diluted plasma samples. Bound antibodies were then detected using a biotin-labeled anti-mouse IgG monoclonal antibody, followed by streptavidin-peroxidase. After the washing steps, a colorimetric reaction was initiated by adding a hydrogen peroxide substrate solution and subsequently stopped with an acidic solution. The absorbance was measured at 450 nm, with a reference measurement at 650 nm, via a multimode microplate reader (CLARIOstar, BMG, Ortenberg, Germany). The corrected absorbance (450–650 nm) was directly proportional to the anti-FIX IgG antibody concentration, which was determined using a standard curve generated from known concentrations of the monoclonal antibody against hFIX (Assaypro LLC, St. Charles, MO, USA).

### 4.9. FIX-Neutralizing Antibody Detection (Bethesda Inhibitor)

FIX-neutralizing antibody levels were determined using a modified Bethesda assay [37]. This assay is based on the aPTT method and is performed on a fully automated blood coagulation analyzer. Briefly, the test plasma (diluted as needed) was mixed 1:1 with standard human plasma and then combined with an equal volume of FIX-deficient human plasma (Affinity Biologicals, Ancaster, ON, Canada). FIX activity was then measured using the aPTT assay. A standard curve was generated through the use of serially diluted human standard plasma (Siemens, Munich, Germany), with activity values assigned according to the manufacturer’s specifications. The residual FIX activity (RA) in the test sample was calculated relative to that of the control. One Bethesda unit (BU) was defined as the amount of inhibitor in 1 mL of plasma that neutralizes 50% of FIX clotting activity. The number of BUs was calculated using the following formula: Bethesda Unit = [2 − log (%Factor IX Residual Activity)]/0.30103. Only RA values between 25% and 75% were used for BU calculations, as values above 75% indicated the absence of an inhibitor. The calculated number of BUs was then multiplied by the plasma dilution factor to obtain the actual inhibitor titer.

### 4.10. Safety Assessments

For urinalysis, urine samples were collected from the mice using metabolic cages over a 16–18 h period prior to study termination. Urine chemistry analysis was performed using a urinalysis analyzer (AE-4020, ARKRAY, Kyoto, Japan) according to the manufacturer’s instructions. For hematology, whole blood collected in K2EDTA tubes (purple-headed vacutainer) was subjected to routine hematological examinations using an automated hematology analyzer (XT-2000IV, Sysmex, Hyogo, Japan) following the manufacturer’s guidelines. For biochemistry, the biochemical parameters of the serum samples collected at study termination were analyzed via automated clinical chemistry analyzers (TBA-25FR and TBA-40FR; TOSHIBA, Tokyo, Japan) according to the manufacturer’s instructions.

### 4.11. Histopathological Examination

At each designated time point and study termination, the mice were euthanized with CO_2_, and gross necropsies were performed. Major organs, including the heart, liver, spleen, lung, kidney, brain, and testis, were collected, weighed, and either stored at −65 to −85 °C for biodistribution assays or fixed in 10% neutral formalin for histopathological analysis. Formalin-fixed tissues were trimmed, dehydrated, paraffin-embedded, and sectioned. Hematoxylin and eosin (H&E) staining was performed, and the slides were examined microscopically (BX53, Olympus, Tokyo, Japan) by a veterinary pathologist. Histopathological observations were quantitatively recorded according to severity grading criteria [38]. The severity grading system for all microscopic lesions was as follows: Minimal severity—less than 10% of the tissue or structure was affected. This grade was used when hyperplasia, hypoplasia, and/or atrophy lesions involved a 10% increase or decrease in volume. Mild severity—10–39% of the tissue or structure was affected. For hyperplasia, hypoplasia, and/or atrophy lesions, this grade was used when the affected structure showed a 10–39% increase or decrease in volume. Moderate severity—40–79% of the tissue or structure was affected. For hyperplasia, hypoplasia, and/or atrophy lesions, this grade was used when the affected structure showed a 40–79% increase or decrease in volume. Marked severity—80–100% of the tissue or structure was affected. For hyperplasia, hypoplasia, and/or atrophy lesions, this grade was used when the affected structure showed an 80–100% increase or decrease in volume. The pathological lesions in the treated groups (therapeutic dose and supraphysiological dose) on Day 2, Day 15, Day 29, and Day 91 were recorded and compared with those in the control group. One-way analysis of variance (ANOVA) followed by an unpaired Student’s *t* test was used to determine significant differences (*p* < 0.05) between the control and treated groups.

### 4.12. Statistical Analysis

All the data are presented as the means ± standard deviations (SDs). Statistical analysis was performed using Student’s *t* test, with a *p* value of ≤0.05 considered statistically significant.

## 5. Conclusions

This study demonstrated that AAV8-FIX-TripleL effectively restored and sustained high FIX activity in hemophilia B mice and had a favorable safety profile. FIX activity was detectable by Day 2 post-injection, with activity peaking at 1149.2% on Day 29 in the therapeutic dose group (5 × 10^11^ VG/kg) and remaining above 800% on Day 91. The plasma FIX levels exceeded 1500 ng/mg from Day 15 to Day 91, with biodistribution analysis confirming predominant liver accumulation, which was consistent with AAV8 tropism. Toxicological evaluations revealed no significant treatment-related adverse effects at doses up to 5 × 10^12^ VG/kg, and the levels of thrombosis markers (TAT and D-dimer) remained comparable to those of the controls. Immunogenicity assessments revealed dose-dependent anti-AAV8 IgG responses, while anti-FIX IgG was found in some mice receiving high doses but declined over time. Overall, our findings support the potential of AAV8-FIX-TripleL as a promising gene therapy candidate for hemophilia B.

## Figures and Tables

**Figure 1 ijms-26-06073-f001:**
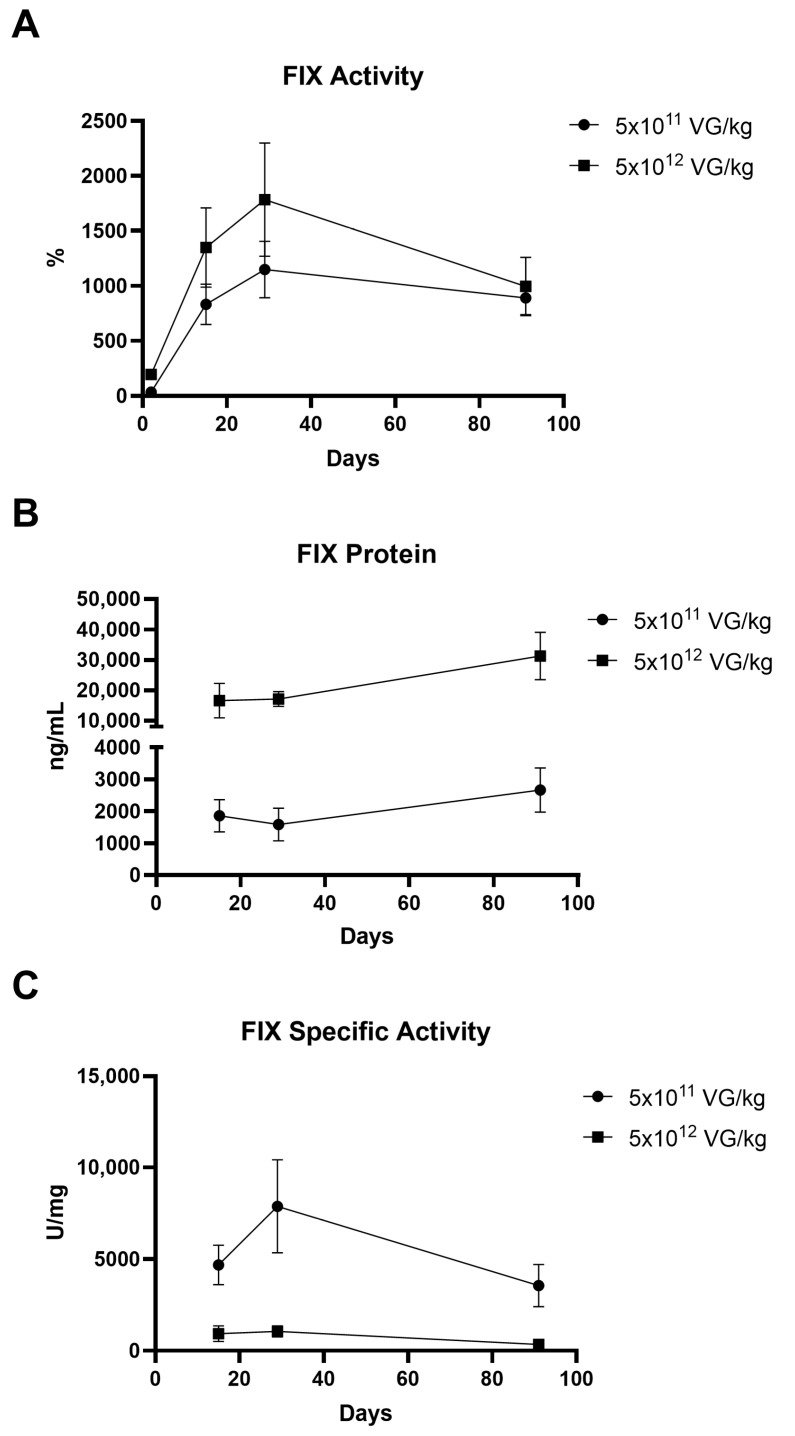
Expression of hFIX following intravenous injection of AAV8-FIX-TripleL in hemophilia B mice. Hemophilia B mice were injected with AAV8-FIX-TripleL at doses of 5 × 10^11^ VG/kg or 5 × 10^12^ VG/kg and sacrificed on Days 2, 15, 29, and 91. Plasma samples were collected to assess FIX activity, protein levels, and specific activity. The data are presented as the means ± SDs (*n* = 15 per group). (**A**) Circulating FIX activity measured using a one-stage clotting assay (FIX-specific activated partial thromboplastin time). (**B**) Total hFIX protein levels detected by ELISA. On Day 2, FIX protein was undetectable in all groups. (**C**) Specific activity of circulating FIX, calculated by dividing FIX activity (U/mL) by the FIX protein concentration (mg/mL). Since FIX protein was undetectable on Day 2, no specific activity data are available for this time point. hFIX, human factor IX; FIX, factor IX; AAV, adeno-associated virus; SD, standard deviation; ELISA, enzyme-linked immunosorbent assay; VG, viral genome. The limit of detection (LOD) for the FIX ELISA was 312.5 ng/mL, and the LOD for FIX activity was 0.625%.

**Figure 2 ijms-26-06073-f002:**
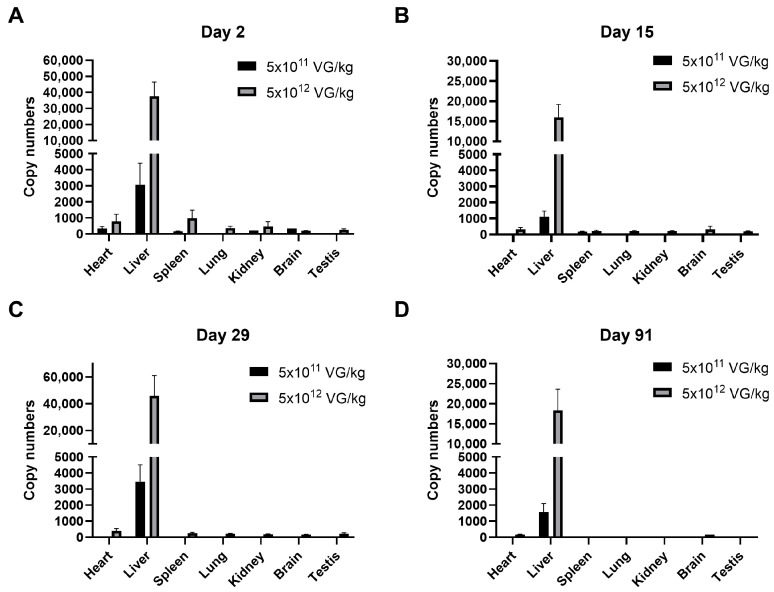
Biodistribution of AAV8-FIX-TripleL following intravenous injection of AAV8-FIX-TripleL in hemophilia B mice. Gene copy numbers of AAV8-FIX-TripleL DNA were assessed in various tissues, including the heart, liver, spleen, lung, kidney, brain, and testis, on Day 2 (**A**), Day 15 (**B**), Day 29 (**C**), and Day 91 (**D**) following intravenous administration at doses of 5 × 10^11^ VG/kg or 5 × 10^12^ VG/kg. AAV8-FIX-TripleL copy numbers are expressed as copies per 10 ng of total DNA. The data are presented as the means ± SDs (*n* = 15 per group). Copy numbers in the control group were undetectable and therefore are not displayed. FIX, factor IX; AAV, adeno-associated virus; SD, standard deviation; VG, viral genome.

**Figure 3 ijms-26-06073-f003:**
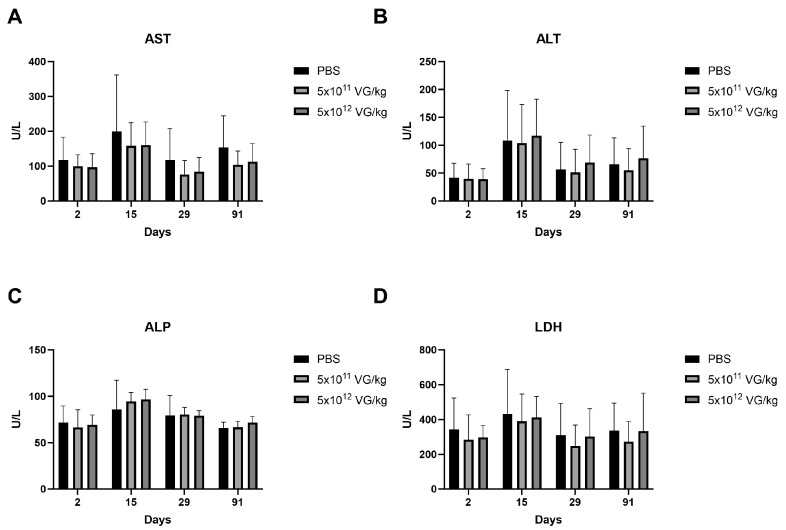
Liver enzyme profiles in the serum of hemophilia B mice following AAV8-FIX-TripleL administration. Serum levels of AST (**A**), ALT (**B**), ALP (**C**), and LDH (**D**) were measured at Days 2, 15, 29, and 91 following intravenous administration of AAV8-FIX-TripleL at doses of 5 × 10^11^ VG/kg or 5 × 10^12^ VG/kg. The data are presented as the means ± SDs (*n* = 15 per group). No statistically significant differences were observed between the dose groups and the control group at any time point. FIX, factor IX; AAV, adeno-associated virus; SD, standard deviation; AST, aspartate aminotransferase; ALT, alanine aminotransferase; ALP, alkaline phosphatase; LDH, lactate dehydrogenase; VG, viral genome.

**Figure 4 ijms-26-06073-f004:**
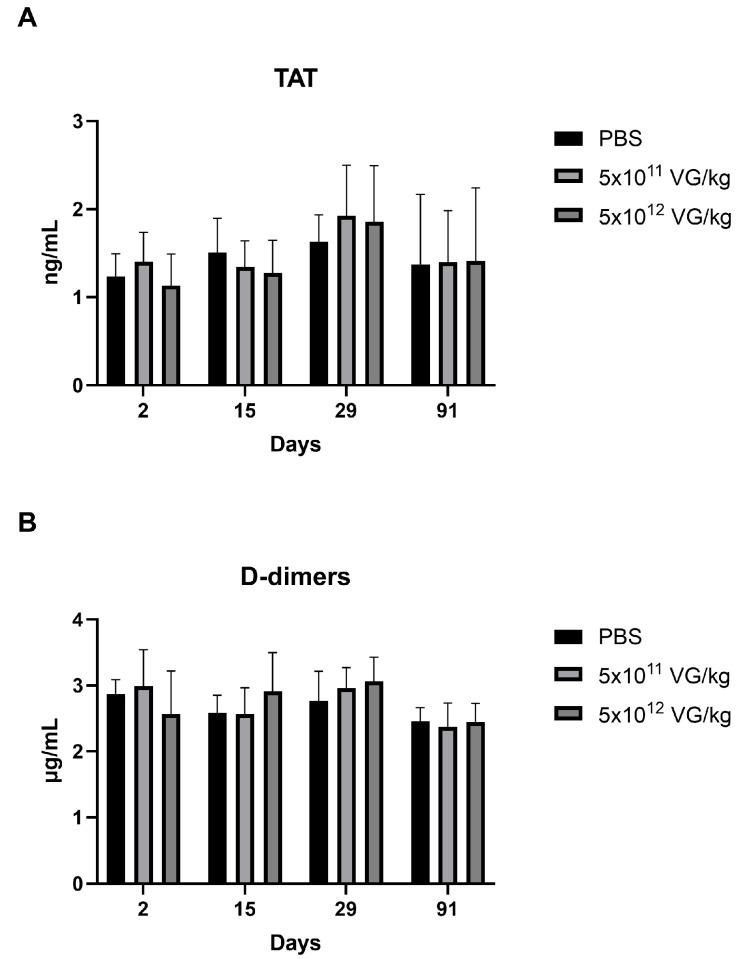
Coagulation activation markers, including thrombin–antithrombin complexes and D-dimers, at various time points following AAV8-FIX-TripleL administration. Plasma levels of TAT (**A**) and D-dimers (**B**) were measured at Days 2, 15, 29, and 91 following intravenous administration of AAV8-FIX-TripleL at doses of 5 × 10^11^ VG/kg or 5 × 10^12^ VG/kg. The data are presented as the means ± SDs (*n* = 15 per group). No statistically significant differences were observed between the dose groups and the control group at any time point. FIX, factor IX; AAV, adeno-associated virus; SD, standard deviation; TAT, thrombin–antithrombin; VG, viral genome.

**Table 1 ijms-26-06073-t001:** Summary of experimental groups, dosages, animal numbers, and study endpoints.

Group	Vector Treatment	Dose (VG/kg)	Harvest Day	Animal Number/Sex
1	PBS	-	Day 2	15/Male
2	AAV8-FIX-TripleL	5 × 10^11^	Day 2	15/Male
3	AAV8-FIX-TripleL	5 × 10^12^	Day 2	15/Male
4	PBS	-	Day 15	15/Male
5	AAV8-FIX-TripleL	5 × 10^11^	Day 15	15/Male
6	AAV8-FIX-TripleL	5 × 10^12^	Day 15	15/Male
7	PBS	-	Day 29	15/Male
8	AAV8-FIX-TripleL	5 × 10^11^	Day 29	15/Male
9	AAV8-FIX-TripleL	5 × 10^12^	Day 29	15/Male
10	PBS	-	Day 91	15/Male
11	AAV8-FIX-TripleL	5 × 10^11^	Day 91	15/Male
12	AAV8-FIX-TripleL	5 × 10^12^	Day 91	15/Male

PBS, phosphate-buffered saline; AAV, adeno-associated virus; FIX, factor IX; VG, viral genome.

**Table 2 ijms-26-06073-t002:** Summary of histopathological findings.

Days		Day 2			Day 15			Day 29			Day 91		
Dose (VG/kg)		-	5 × 10^11^	5 × 10^12^	-	5 × 10^11^	5 × 10^12^	-	5 × 10^11^	5 × 10^12^	-	5 × 10^11^	5 × 10^12^
Lung	Osseous metaplasia, alveoli, focal, minimal	1/15	-	-	-	-	-	-	-	-	-	-	-
	Score value	0.07 ± 0.26	-	-	-	-	-	-	-	-	-	-	-
	Infiltration, mononuclear cell, focal, minimal	-	-	1/15	-	-	1/15	-	-	1/15	1/15	-	1/15
	Score value	-	-	0.07 ± 0.26	-	-	0.07 ± 0.26	-	-	0.07 ± 0.26	0.07 ± 0.26	-	0.07 ± 0.26
Heart	Infiltration, mononuclear cell, focal, minimal	1/15	-	-	-	-	-	1/15	-	-	-	-	-
	Score value	0.07 ± 0.26	-	-	-	-	-	0.07 ± 0.26	-	-	-	-	-
	Mineralization, focal, minimal	-	-	-	1/15	-	-	-	-	-	-	-	-
	Score value	-	-	-	0.07 ± 0.26	-	-	-	-	-	-	-	-
	Cardiomyopathy, focal, minimal	-	-	-	-	-	-	-	-	1/15	-	-	-
	Score value	-	-	-	-	-	-	-	-	0.07 ± 0.26	-	-	-
Spleen	Pigment, focal, minimal	1/15	1/15	-	-	-	-	-	-	-	-	-	-
	Score value	0.07 ± 0.26	0.07 ± 0.26	-	-	-	-	-	-	-	-	-	-
Pancreas	Vacuolation, focal, minimal	1/15	-	-	-	-	-	-	-	-	-	-	-
	Score value	0.07 ± 0.26	-	-	-	-	-	-	-	-	-	-	-
Liver	Infiltration, mononuclear cell, focal, minimal	8/15	5/15	7/15	9/15	9/15	6/15	5/15	9/15	8/15	8/15	7/15	2/15
	Score value	0.53 ± 0.52	0.33 ± 0.49	0.47 ± 0.52	0.60 ± 0.51	0.60 ± 0.51	0.40 ± 0.51	0.33 ± 0.49	0.60 ± 0.51	0.60 ± 0.63	0.53 ± 0.52	0.47 ± 0.52	0.13 ± 0.35 *
	Necrosis, focal, minimal	-	-	-	2/15	-	-	-	-	-	-	-	-
	Score value	-	-	-	0.13 ± 0.35	-	-	-	-	-	-	-	-
	Fibrosis, focal, mild	-	-	-	-	-	-	-	-	1/15	-	-	-
	Score value	-	-	-	-	-	-	-	-	0.13 ± 0.52	-	-	-
Kidneys	Infiltration, mononuclear cell, cortex, focal, minimal	1/15	3/15	1/15	3/15	-	3/15	4/15	-	3/15	4/15	3/15	5/15
	Score value	0.07 ± 0.26	0.20 ± 0.41	0.07 ± 0.26	0.20 ± 0.41	-	0.20 ± 0.41	0.27 ± 0.46	-	0.20 ± 0.41	0.27 ± 0.46	0.27 ± 0.59	0.33 ± 0.49
	Vacuolation, glomerulus, focal, minimal	-	-	1/15	-	-	-	-	-	-	-	-	-
	Score value	-	-	0.07 ± 0.26	-	-	-	-	-	-	-	-	-
	Cyst, cortex/medulla, focal, minimal to moderate	3/15	-	-	-	1/15	-	-	-	-	-	-	-
	Score value	0.40 ± 0.91	-	-	-	0.07 ± 0.26	-	-	-	-	-	-	-
	Cast, renal tubule, cortex, focal, minimal	2/15	-	-	-	1/15	-	1/15	1/15	-	-	1/15	2/15
	Score value	0.13 ± 0.35	-	-	-	0.07 ± 0.26	-	0.07 ± 0.26	0.07 ± 0.26	-	-	0.07 ± 0.26	0.13 ± 0.35
	Basophilia, tubule, cortex, focal, minimal	-	-	-	-	1/15	-	-	-	-	-	-	-
	Score value	-	-	-	-	0.07 ± 0.26	-	-	-	-	-	-	-
	Dilation, tubule, cortex, focal, minimal	-	-	-	-	-	-	1/15	-	-	-	-	-
	Score value	-	-	-	-	-	-	0.07 ± 0.26	-	-	-	-	-
Adrenals	Hyperplasia, subcapsular, cortex, focal, minimal	2/15	1/15	1/15	-	1/15	-	1/15	-	-	-	2/15	-
	Score value	0.13 ± 0.35	0.07 ± 0.26	0.07 ± 0.26	-	0.07 ± 0.26	-	0.07 ± 0.26	-	-	-	0.13 ± 0.35	-
	Accessory adrenocortical nodule, focal, minimal to mild	-	2/15	-	-	-	-	-	-	-	1/15	1/15	-
	Score value	-	0.20 ± 0.56	-	-	-	-	-	-	-	0.07 ± 0.26	0.07 ± 0.26	
Testis	Degeneration, germ cell, focal, moderate	-	-	-	-	-	-	1/15	1/15	-	-	-	-
	Score value	-	-	-	-	-	-	0.20 ± 0.77	0.20 ± 0.77	-	-	-	-

*, *p* < 0.05 (compared to control group; unpaired Student’s *t*-test).

**Table 3 ijms-26-06073-t003:** Summary of anti-AAV8 IgG, anti-FIX IgG, and Bethesda inhibitor analysis findings.

Days	Day 2			Day 15			Day 29			Day 91		
Dose (VG/kg)	-	5 × 10^11^	5 × 10^12^	-	5 × 10^11^	5 × 10^12^	-	5 × 10^11^	5 × 10^12^	-	5 × 10^11^	5 × 10^12^
AAV8 antibody/total no. of animals	0/15	0/15	0/15	0/15	15/15	15/15	0/15	15/15	15/15	0/15	15/15	15/15
Range (μg/mL)	-	-	-	-	5.21 ± 2.38	10.99 ± 4.51	-	11.70 ± 5.51	32.79 ± 12.10	-	24.78 ± 25.15	42.50 ± 22.36
FIX antibody/total no. of animals	0/15	0/15	0/15	0/15	1/15	0/15	0/15	0/15	4/15	0/15	0/15	4/15
Range (μg/mL)	-	-	-	-	7.61	-	-	-	3.60 ± 5.48	-	-	0.57 ± 0.31
Bethesda inhibitor/total no. of animals	0/15	0/15	0/15	0/15	0/15	0/15	0/15	0/15	0/15	0/15	0/15	0/15
Titer (BIU)	-	-	-	-	-	-	-	-	-	-	-	-

BIU, Bethesda inhibitor units; FIX, factor IX; AAV, adeno-associated virus; VG, viral genome. Note: anti-AAV8 and anti-FIX antibodies were measured using ELISA-based methods.

## Data Availability

The original contributions presented in this study are included in the article/Appendix A. Further inquiries can be directed to the corresponding authors.

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
