# Peer review of "Preclinical Evaluation of the Systemic Safety, Efficacy, and Biodistribution of a Recombinant AAV8 Vector Expressing FIX-TripleL in Hemophilia B Mice: Implications for Human Gene Therapy"

_ijms, 2025, doi:10.3390/ijms26136073_

Round 1

Reviewer 1 Report

Comments and Suggestions for Authors

This is a well-written paper describing the use of liver-directed AAV8 gene therapy using an improved liver-targeting vector for the treatment of factor IX deficiency. I only have few comments. 

Table 1 should be referenced in the beginning of section 2.1 of the results for clarity. 

The word symptoms should be replaced with signs in animals (symptoms is only used for humans). 

Figure 1C: I am not sure I understand why there would be less specific activity at the higher dose?

In the legend for Figure 1, it is stated: "Since FIX protein was undetectable on Day 2, no specific activity data are available for this time point." But yet 0 activity is still data and should be noted in the graph. 

Sometimes in the text, the authors state that the control group received vehicle and other times it says PBS. I would think that vehicle is empty vector. Is this incorrect? 

Reviewer 2 Report

Comments and Suggestions for Authors

This manuscript investigates the systemic safety, efficacy, biodistribution, and immunogenicity of AAV8-FIX-tripleL administered via intravenous at therapeutic and supraphysiological doses in FIX-knockout hemophilia B (HB) mice. In general, the manuscript holds significant value for validating the efficacy and safety of high-activity FIX-tripleL in this model. However, it suffers from several limitations that are amenable to revision. Major revision is recommended before it can be accepted. Specific comments include the following:

  1. Thrombosis risk assessment at therapeutic and supraphysiological doses requires enhancement.When using gene therapy to treat HB, restoring FIX activity to 5–10% is sufficient to alleviate bleeding symptoms. Whether higher FIX activity is always better or if an optimal activity range is required remains unclear. The FIX-tripleL sequence, derived from the highly active FIX-R338L (FIX-Padua) and incorporating two additional mutations (V86A/E277A), exhibits a specific activity 2.9-fold higher than FIX-Padua. Given that elevated FIX activity correlates with stronger coagulation function, the risk of thrombosis, particularly venous thromboembolism (VTE), also increases, as a hypercoagulable state is one of the mechanisms underlying pathogenesis. Considering the use of this hyperactive FIX variant at therapeutic and supraphysiological doses, a comprehensive assessment of coagulation function is required. Detection of TAT and D-dimer levels is insufficiently persuasive. It is essential to include measurements of thrombin generation/activity and fibrinogen levels. Furthermore, evaluating the risk of VTE directly, potentially using a venous thrombosis model, is strongly recommended to provide convincing evidence for the safety at therapeutic and supraphysiological
  2. Since the purpose of this study is to evaluate the efficacy and safety of high-activity FIX-tripleL, the advantage of high activity lies in reducing the viral vector dose. However, this study did not use low and extremely low doses to assess its efficacy in alleviating bleeding. If the efficacy of low or extremely low doses is comparable to that of FIX WT or FIX-R338L at therapeutic doses, it would provide stronger evidence of the advantages of high-activity FIX-tripleL. Therefore, the authors are strongly encouraged to include efficacy data at low and extremely low doses.
  3. The authors selected four time points in the study: Days 2, 15, 28, and 91. Please clarify the rationale for selecting these four time points.Based on the results in Figure 1A, even at the endpoint of Day 91, FIX activity remains as high as almost 1000%. Such high activity is sufficient for produce effective therapeutic effect. To determine the duration of therapeutic efficacy for both therapeutic and supraphysiological doses, it is recommended to extend the observation period significantly beyond 91 days. Additionally, Figure 1A suggests that by Day 91, FIX activity levels in the supraphysiological dose group are comparable to those in the therapeutic dose group. Does this indicate that the supraphysiological dose offers no significant long-term therapeutic advantage over the therapeutic dose?
  4. In the histopathology analysis results, Table 2 only describes some of the pathological observations, the original hematoxylin and eosin (H&E) stained images supporting these findings are not provided. Inclusion of representative original images is essential to enhance the credibility and persuasiveness of the histopathology results.
  5. Regarding the assessment of subcutaneous hemorrhage in HB mice by gross necropsy and the improvement of the subcutaneous hemorrhage after gene therapy, it is also recommended to provide the original photographs to enhance the persuasiveness of the finding.

Round 2

Reviewer 2 Report

Comments and Suggestions for Authors

I have carefully reviewed the revisions made by the authors in response to my previous comments. I am pleased to note that the authors have addressed nearly all of my concerns. They added the data from the pilot study at the low-dose level to evaluate the efficacy of the FIX-tripleL. They also provided some original H&E-stained images supporting the histopathology analysis in Supplementary Fig.4, and the representative gross necropsy images demonstrating subcutaneous hemorrhage in Supplementary Fig.3. In terms of safety evaluation, the authors explained the practical reasons for not testing fibrinogen levels and TGA, which are generally understandable and acceptable. In general, the revisions have significantly improved the clarity, accuracy, and overall quality of the manuscript. I have no further comments or suggestions for revision.